# Beyond Borders of the Cell: How Extracellular Vesicles Shape COVID-19 for People with Cystic Fibrosis

**DOI:** 10.3390/ijms25073713

**Published:** 2024-03-27

**Authors:** Ewelina D. Hejenkowska, Hayrettin Yavuz, Agnieszka Swiatecka-Urban

**Affiliations:** Department of Pediatrics, University of Virginia, Charlottesville, VA 22903, USA; stp3bt@virginia.edu (E.D.H.); eeu9ng@virginia.edu (H.Y.)

**Keywords:** COVID-19, cystic fibrosis, extracellular vesicles, TGF-β

## Abstract

The interaction between extracellular vesicles (EVs) and SARS-CoV-2, the virus causing COVID-19, especially in people with cystic fibrosis (PwCF) is insufficiently studied. EVs are small membrane-bound particles involved in cell–cell communications in different physiological and pathological conditions, including inflammation and infection. The CF airway cells release EVs that differ from those released by healthy cells and may play an intriguing role in regulating the inflammatory response to SARS-CoV-2. On the one hand, EVs may activate neutrophils and exacerbate inflammation. On the other hand, EVs may block IL-6, a pro-inflammatory cytokine associated with severe COVID-19, and protect PwCF from adverse outcomes. EVs are regulated by TGF-β signaling, essential in different disease states, including COVID-19. Here, we review the knowledge, identify the gaps in understanding, and suggest future research directions to elucidate the role of EVs in PwCF during COVID-19.

## 1. Introduction

The coronavirus disease 2019 (COVID-19) pandemic has affected over 773 million individuals, and it resulted in the deaths of more than 6.9 million people between December 2019 and 2023 [1,2]. The enveloped positive-sense single-stranded RNA severe acute respiratory syndrome coronavirus (SARS-CoV)-2 is the cause of COVID-19 [3]. The spike (S) glycoprotein homotrimer on the SARS-CoV-2 envelope interacts with angiotensin-converting enzyme 2 (ACE2) on the surface of the host airway epithelial cells [4,5]. Human proteases such as transmembrane serine protease 2 (TMPRSS2), mosaic serine protease (TMPRSS13), and human airway trypsin-like protease (TMPRSS11D) activate each S monomer upon binding to ACE2 and cleave them into S1 and S2 subunits [3,4,6,7,8,9]. In particular, the S1 subunit’s receptor binding domain (RBD) binds to the human ACE2 [3]. Hence, these cells are the port of entry for SARS-CoV-2 [10]. It has been demonstrated by in vitro and in vivo experiments that downregulating *ACE2* expression decreases SARS-CoV-2 infection [11]. Additionally, it was found that transforming growth factor-beta 1 (TGF-β1) decreases ACE2 protein abundance through the microRNA-mediated mechanism and may decrease RBD binding to ACE2 [12]. ACE2 plays a crucial anti-inflammatory and anti-fibrotic role by converting angiotensin (Ang) I to Ang 1–9 and Ang II to Ang 1–7 [10]. Although the lower density of the cell surface of ACE2 may hinder entry of the viral particle, it may also increase inflammation and fibrosis because of the decreased conversion of Ang I and Ang II. Accordingly, the lower cell surface density of ACE2 is associated with increased Ang II levels and may contribute to acute lung injury and fibrosis during COVID-19 [13,14,15].

People with cystic fibrosis (PwCF) often experience severe respiratory illness aggravated by bacterial and viral infections [16]. It has been proposed that PwCF may have a milder course of the disease and may be less susceptible to SARS-CoV-2 infection than the general population if they do not suffer from pre-existing severe lung illness [17,18,19,20,21,22]. Attention to universal infectious precautions, self-distancing, and certain medications were thought to be responsible for less severe COVID-19 in PwCF [20,23]. For instance, azithromycin, which is frequently used to treat bacterial infections in PwCF, may reduce SARS-CoV-2 infection and COVID-19 severity; however, it did not prevent symptoms of COVID-19 in the non-CF population when compared to a placebo [22,24]. CFTR modulators, a new class of drugs increasing CFTR function in PwCF, decreased SARS-CoV-2 replication in a primary CF airway cell model [25]. CFTR modulators were also associated with a significant decrease in hospitalization of PwCF requiring oxygen, and it was concluded that having lower lung function is linked to more severe outcomes in COVID-19 [26]. Other investigators found a higher incidence of SARS-CoV-2 infection in PwCF despite reports of less severe COVID-19 in this population [27,28]. Another study suggested that CFTR may facilitate the virus replication because IOWH-032, a small molecule CFTR inhibitor, suppressed SARS-CoV-2 replication [29]. IOWH-032 is a synthetic extracellular CFTR inhibitor that entered a phase II clinical trial in 2013 to treat diarrhea but has not progressed to clinical development. It was later found to have a potentiating effect on human CFTR [30]. Based on the newer information that IOWH-032 is an ortholog-specific CFTR inhibitor and potentiator, it cannot be concluded that CFTR aids in SARS-CoV-2 replication. Overall, the studies discussed above do not offer a consensus on whether CFTR dysfunction or CF-directed therapies modify the risk of SARS-CoV-2 infection or the severity of COVID-19 in PwCF. It is unknown whether CFTR modulators or CFTR expression or function influence the SARS-CoV-2 viral load.

Some mechanistic cues about the role of CFTR in SARS-CoV-2 infection have been provided by the following studies. First, CFTR was shown to colocalize with the SARS-CoV-2 receptor, ACE2, in several types of epithelial cells, including those in the respiratory tract [31]. Second, the SARS-CoV-2 nucleocapsid was proposed to downregulate *CFTR* expression [32]. These findings suggest that the nucleocapsid-dependent loss of CFTR function may result from ACE2 interaction with CFTR, thus initiating a CF-like phenotype and inducing an inflammatory response similar to one experienced by PwCF. It was shown that the S-protein may cause host cell inflammation through TLR2-dependent activation of the NF-κB pathway [33]. Thus, ACE2’s anti-inflammatory properties may counteract the S-protein-induced inflammation. Indeed, non-CF and CF airway epithelial cells show varying levels of ACE2 in studies, with opposing results suggesting other factors contributing to the infection rate and severity of COVID-19 [12,25,31,34,35,36]. In recent years, several studies have examined the role of extracellular vesicles (EVs) in host–pathogen interactions and host cell communications in the pathogenesis of tissue inflammation and fibrosis in different disease states, including chronic respiratory diseases, SARS-CoV-2 infection, and CF pathogenesis [37,38,39,40]. Respiratory viruses use the host EVs to modulate their transmission. Rhinoviruses induce the release of EVs from airway epithelial cells that stimulate viral receptor expression on monocytes and turn uninfected cells into more permissive ones [41,42]. The influenza H5N1 virus induces the release of EVs from infected cells that trigger inflammation [43]. We review the present knowledge of how SARS-CoV-2 influences the release and content of host EVs and how the EVs modulate COVID-19 severity in PwCF.

## 2. The Origin and Role of Extracellular Vesicles (EVs)

EVs are lipid bilayer-enclosed nanoparticles released into the extracellular space by living cells, including prokaryotes [44,45]. In 1945, Chargaff discovered a sediment with the capability of shortening clotting, and a year later Chargaff and West discovered that this fraction—later identified as EVs—had a high potential of clotting [46,47]. In 1967, Peter Wolf discovered particles distinguishable from platelets by electron microscopy, described as ‘platelet dust’ [48]. In 1971, it was demonstrated that EVs had cargo such as contractile proteins, and in 1989, it was shown that EVs were enzymatically active [49,50]. Until the 1990s, EVs were described as “trash cans/garbage bins” [51,52]. In 1996, Raposo et al. suggested that EVs play a role in antigen presentation in vivo, and they proved that EVs are biologically functional [53,54].

Various nomenclatures have been used for nanoparticles based on their origin, cargo, and size. Based on the diameter, particles smaller than 200 nm in diameter are described as small EVs, while particles larger than 200 nm in diameter are described as large EVs [44]. Based on the cellular origin, larger-sized particles originating from the plasma membrane are called microvesicles, smaller-sized particles originating from the endosomal system are called exosomes, and larger-sized particles originating from the plasma membrane are called ectosomes. There are many other subgroups of EVs such as large oncosomes, apoptotic bodies, migrasomes, and exosome-like vesicles [44,55]. All of these are considered subgroups of EVs, and “EV” is currently used as the generic term, according to the minimal information for studies of extracellular vesicles (MISEV 2023) guidelines [44,56,57].

Each EV subtype has specific membrane markers that provide information about its origin and are important for identification and detection, e.g., markers of exosomes are tetraspanins (CD9, CD63, CD81), markers of microvesicles are Annexin A1, selectins, and integrins, and markers of apoptotic bodies formed by cellular blebbing from the cell in the apoptosis process are Caspase 3 and Annexin V [55].

EVs can be isolated from almost all body fluids including urine, blood, cerebrospinal fluid, and bronchoalveolar lavage fluid (BALF) [58]. EVs play an essential role in intercellular communication and carry a cargo of proteins, lipids, and genetic material such as DNA, mRNA, and microRNA derived from parental cells [59,60]. The protein cargo of EVs is related to the cellular origin and the mechanism of biogenesis. For example, EVs originating from the endolysosomal compartment are enriched in major histocompatibility complex class II (MHC Class II) and tetraspanins, while EVs originating from the plasma membrane are enriched in integrins, glycoprotein 1-b, and P-selectin. Apoptotic bodies contain apoptosis-associated proteins such as histones [61,62]. Furthermore, EVs provide intercellular communication between donor and recipient cells by releasing them and taking them up [63]. Furthermore, they are involved in both physiological and pathological processes such as inflammation, immune response, antigen presentation, and cancer development by delivering cargo to recipient cells [64,65].

## 3. The Role of EVs in the Pathogenesis of Inflammation

EVs are known to play a role in local and systemic inflammation [39]. EVs originating from the lung epithelial and endothelial cells, alveolar macrophages, and neutrophils play a role in the pathogenesis of chronic respiratory diseases [37,66,67,68]. During infections, such as those caused by SARS-CoV-2, alveolar macrophages are believed to be the primary source of EVs in BALF [69]. In cancer, EVs may contain high levels of TGF-β1, along with other molecules like mRNAs and proteins linked to TGF-β signaling [70]. The cell–cell communications mediated by EVs can intensify the inflammatory response and cellular damage [70,71]. EVs have been shown to modulate the host response to different viruses, including SARS-CoV-2 [72,73]. For example, breast milk-derived EVs are able to inhibit HIV-1 infection of monocyte-derived dendritic cells and block viral transfer to CD4+ T cells [74]. EVs have been shown to decrease COVID-19 severity. Blocking pro-inflammatory cytokine IL-6 and upregulating the anti-inflammatory IL-10 reduces viral replication and decreases the systemic inflammation associated with COVID-19 [72,75,76]. However, little is known about the role of EVs in PwCF with COVID-19. Therefore, it is important to review the current literature on EVs in PwCF with COVID-19 to identify the potential mechanisms and implications of EV-mediated interactions between the virus and the host, gain a deeper understanding of their function, identify the knowledge gaps, and suggest future research directions.

## 4. The Role of EVs in the Pathogenesis of CF

EVs may play an important role in the pathogenesis of chronic lung disease, including CF [68]. In the CF patient’s airway, EVs produced by epithelial cells differ from those generated by healthy cells in their protein content. For example, EVs derived from CFBE41o-, a human bronchial epithelial cell line from a CF patient, showed a significant increase in proteins associated with acute inflammation and infection, such as vascular cell adhesion protein 1 (VCAM1) and S100 calcium-binding protein A12 (S100A12) [38]. In PwCF, BALF-derived EVs were enriched in grancalcin and histones [77]. These findings may influence neutrophils, driving their recruitment [78,79,80]. Recently, it was demonstrated that CF-derived EVs may activate neutrophils, causing the release of Caspase-1 and Interleukin (IL)-1 [81]. As shown in Table 1, research has advanced our understanding of EVs’ cargo in PwCF. Still, studies have yet to investigate how EVs influence the recruitment and activation of neutrophils. Many CFTR mutations cause CF and present with variable severity, and the cargo and effects of EVs likely differ among PwCF. Therefore, further studies are necessary to comprehend this diversity and what it means for COVID-19 prevention and therapeutic strategies in this patient population.

The breakdown of neutrophil-derived serine proteases, IL-6, was suggested as a cause of less severe COVID-19 in PwCF [23]. EVs released from airway cells in PwCF may block IL-6-induced synthesis of acute phase proteins, resulting in less inflammation and less severe COVID-19. Indeed, it has been shown that higher expression of *IL-6* is seen in severe cases of COVID-19 [85]. However, other studies showed that SARS-CoV-2 seroconversion in people with chronic kidney disease is promoted by pro-inflammatory cytokines IL-6 and IFN-γ [86]. Patients with chronic kidney disease who were either infected with or vaccinated against SARS-CoV-2 had higher levels of IL-6 and IFN-γ, compared to healthy unvaccinated individuals [86]. Thus, EVs might block IL-6, which stops IL-6-induced synthesis of acute phase proteins. This could serve as a protective measure and potentially decrease the severity of COVID-19 (Figure 1) [86]. Although this study examined only patients with kidney disease, it may be possible that EV-induced blockage of IL-6 arises in PwCF infected with SARS-CoV-2, since inflammation is central to the pathogenesis of CF and chronic kidney disease [87,88].

## 5. EVs As Mediators of SARS-CoV-2 Infection

Several studies have been conducted to determine the role of EVs in SARS-CoV-2 pathogenesis. In these studies, SARS-CoV-2 caused the release of EVs from several types of host cells, including platelets [89], lung epithelial cells A549 [90], and Vero E6 cells [91]. Detection of EVs and their cargo in serum may serve as a prognostic indicator of the severity of COVID-19 [92,93]. Numerous molecules implicated in the immunological response, inflammation, and activation of the coagulation and complement pathways were discovered through proteomic analysis of patient-derived circulating EVs, causing multi-organ dysfunctions linked to COVID-19 [94]. It is still unclear how activating the inflammatory, complement, and coagulation pathways during COVID-19 contributes to the disease progression and severity in PwCF. The number of EVs increases during respiratory viral infection in PwCF [95]. According to a recent study, EVs secreted by various SARS-CoV-2-infected cells contain large amounts of live virus particles [91]. Thus, SARS-CoV-2 could evade neutralizing antibodies through EV-mediated cell-to-cell transmission. It has also been reported that EVs play an immunomodulatory role in the recovery from COVID-19 by regulating the functions of CD4+ and CD8+ T lymphocytes [96]. EVs from COVID-19 patients were shown to have ACE2 receptors on their surface, and by contesting with the cellular ACE2’s binding site, these vesicles can function as decoys preventing SARS-CoV-2 infection in vivo [97]. Engineered soluble ACE2-loaded EVs are therapeutically effective against SARS-CoV-2 infection in mice [98]. Several studies focused on the presence of ACE2 in EVs, given that ACE2 is essential for the fusion of SARS-CoV-2 virus particles with the host cell membrane. It was shown that ACE2 was transferred between different cell types via EVs [99]. However, more research is needed to understand the cell–cell transfer of ACE2 via EVs to understand how this process modulates the severity of SARS-CoV-2 infection. EVs can also play a dual role in SARS-CoV-2 pathogenesis. Similar to CF pathogenesis, neutrophils are activated upon SARS-CoV-2 viral infection [100]. EVs could activate neutrophils, which might also trigger the release of serine proteases, which are believed to result in less severe COVID-19 by breaking down IL-6 [23,36]. However, it was shown that higher levels of IL-6 in SARS-CoV-2-infected people may lead to an improved seroconversion rate [86].

## 6. The Role of EVs in TGF-β Signaling

During SARS-CoV-2 infection, TGF-β signaling plays a crucial role and is particularly prominent in the chronic immune response observed in severe COVID-19 cases [101]. Locally produced TGF-β may contribute to pulmonary fibrosis during the later stages of the immune response [101,102,103,104]. Additionally, in PwCF who are homozygous for the most common *cftr* gene mutation F508del, the *TGF-β* gene has been confirmed as a modifier of lung disease severity. Specifically, certain polymorphisms linked to elevated *TGF-β1* expression are associated with more severe lung disease and may impact up to 40% of F508del homozygous patients [105,106,107]. TGF-β signaling induces the secretion of EVs from cancer cells, where the vesicles evoke endothelial barrier instability by facilitating the endothelial–mesenchymal transition (EMT) [108]. Oral squamous cell carcinoma cells that were exposed to TGF-β demonstrated increased release of EVs and altered content, compared to EVs released from untreated cells. The EVs released from cells undergoing EMT after TGF-β treatment demonstrated a lower expression of endothelial cell markers and an increased expression of mesenchymal cell markers [108]. The role of EVs in lung fibrosis is largely unexplored. However, one study suggested that a higher content of Programmed Death-Ligand 1 (PD-L1) in EVs released from fibroblasts upon TGFβ stimulation could contribute to immune suppression and fibrogenesis [109].

## 7. Conclusions

In summary, EVs play a significant role in SARS-CoV-2 pathogenesis, with their number increasing during infection and potentially carrying live virus particles [38]. EVs modulate the severity of SARS-CoV-2 infection in PwCF by modifying cytokines expression (Figure 1). EVs can modulate the immune response, acting as decoys to prevent SARS-CoV-2 infection by contesting with cellular ACE2 receptors, and engineered ACE2-loaded EVs are effective against SARS-CoV-2 infection in mice. TGF-β-induced EVs from cancer cells play a critical role in vascular destabilization. These EVs, secreted during EMT, target endothelial cells and contribute to immune suppression and fibrogenesis. Inflammation is a significant component of CF pathogenesis and SARS-CoV-2 infection [110]. The content of EVs in PwCF and SARS-CoV-2 infection is summarized in Figure 1. For instance, EVs derived from the sputum of patients infected with SARS-CoV-2 have been found to express high levels of specific immune-related proteins, which correlate strongly with the expression of the SARS-CoV-2 N protein [111].

## 8. Knowledge Gaps and Future Directions

Currently, there are no specific treatments for COVID-19 in PwCF, and it is unclear whether the CFTR-directed therapy has any effect. Future research should focus on helping to personalize therapy and reduce the severity of SARS-CoV-2 infections in this vulnerable population. EVs released by cells in PwCF contain proteins that may impact neutrophil recruitment and activation, potentially resulting in less severe COVID-19 by inhibiting IL-6-induced acute phase mediators. More research is needed to elucidate how the EVs influence neutrophil recruitment and activation. More research is also needed to understand how specific *cftr* gene mutations and disease severity influence the EV’s cargo, affecting COVID-19 severity among PwCF.

## Figures and Tables

**Figure 1 ijms-25-03713-f001:**
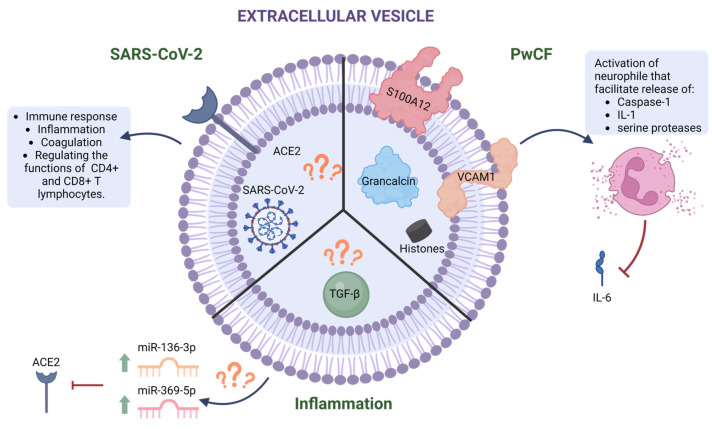
Diagram of EVs’ cargo under different conditions. During SARS-CoV-2 infection, the EVs’ cargo may include ACE2 and, in some instances, fully viable virus particles. This can trigger immune responses, inflammation, coagulation, and regulation of CD4+ and CD8+ T lymphocytes. During the pathogenesis of CF, the cargo is enriched with proteins such as S100A12, grancalcin, VCAM1, and histones. These proteins can activate neutrophils, leading to the release of Caspase-1, IL-1, and serine proteases, which may facilitate the breakage of IL-6. During inflammation, the cargo may include TGF-β ligand, which, when released, can lead to the upregulated expression of miR-136-3p and miR-369-5p, resulting in the downregulation of the ACE2 receptor. These three conditions can interact and influence the pathogenesis of SARS-CoV-2 in PwCF. Created with BioRender.

**Table 1 ijms-25-03713-t001:** The content of EVs isolated from the CF models of human airway cells or BALF in PwCF and controls without CF.

Controls	PwCF
BALF: HSP70; HSP90 [82]	BALF: HSP70; HSP90 [82]
BALF: MHC-I MHC-II [82]	BALF: MHC-I MHC-II [82]
Cells: Metabolites [83] BALF: miR-122-5p; miR-423-5p; miR-375-3p; miR-200a-3p; miR-200b-3p; miR-141-3p [82,84]	Cells: VCAM1 [38] Cells: S100A12 [38] BALF: Grancalcin [77] BALF: Histones [77]

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
