# Peer review of "Beyond Borders of the Cell: How Extracellular Vesicles Shape COVID-19 for People with Cystic Fibrosis"

_ijms, 2024, doi:10.3390/ijms25073713_

Round 1

Reviewer 1 Report

Comments and Suggestions for Authors

This insightful review discusses the interaction between extracellular vesicles (EVs) and the SARS-CoV-2 virus, focusing on individuals with cystic fibrosis (PwCF). It examines the role of EVs in the inflammatory response to the virus, exploring potential both positive and negative effects. Additionally, the involvement of EVs in the pathogenesis of cystic fibrosis and the severity of COVID-19 in individuals affected by this condition is discussed. The text identifies gaps in current understanding and suggests future directions for research.

The paper is generally well written and structured. The abstract is detailed enough for the reader to comprehend the article without having to read the entire manuscript. There is no discrepancy between the abstract and the manuscript. The introduction provides sufficient background and include all relevant references. The discussion is well-written and articulated.

Below few observations:

- The Introduction provides a concise overview of the COVID-19 pandemic, the responsible virus, and the importance of examining the interaction between extracellular vesicles (EVs) and SARS-CoV-2 in people with cystic fibrosis (PwCF). However, it could have been clearer in presenting a well-defined hypothesis or research question to guide the study, thus strengthening the introduction and indicating specific study objectives to the readers.

- The text does not include a "conclusions" section. This section could summarize the main findings, discuss their implications, and highlight any recommendations for future research. Adding it would improve the coherence and completeness of the work.

- To expedite the reading process of the manuscript, include a graphical abstract.

- Improve the resolution of all figures.

- The English is understandable and well-structured, and the grammar is generally correct. However, there are some minor corrections that could be made to improve the fluency and clarity of the text. Some sentences could be rephrased to make the meaning more precise or to avoid repetitions, such as lines "22-23" "The COVID-19 pandemic has afflicted more than 773 million people and killed more than 6.9 million individuals between December 2019 and 2023," which could be rephrased: "The COVID-19 pandemic has affected over 773 million individuals and resulted in the deaths of more than 6.9 million people between December 2019 and 2023." These rephrasings could make the text clearer and avoid excessive use of similar words. Additionally, there are some punctuation errors and sentence structure issues that could be corrected to make the text more coherent. Overall, it's quite good, but there's always room for improvement.

Comments on the Quality of English Language

The English is understandable and well-structured, and the grammar is generally correct. However, there are some minor corrections that could be made to improve the fluency and clarity of the text. Some sentences could be rephrased to make the meaning more precise or to avoid repetitions, such as lines "22-23" "The COVID-19 pandemic has afflicted more than 773 million people and killed more than 6.9 million individuals between December 2019 and 2023," which could be rephrased: "The COVID-19 pandemic has affected over 773 million individuals and resulted in the deaths of more than 6.9 million people between December 2019 and 2023." These rephrasings could make the text clearer and avoid excessive use of similar words. Additionally, there are some punctuation errors and sentence structure issues that could be corrected to make the text more coherent. Overall, it's quite good, but there's always room for improvement.

Author Response

  1. The Introduction provides a concise overview of the COVID-19 pandemic, the responsible virus, and the importance of examining the interaction between extracellular vesicles (EVs) and SARS-CoV-2 in people with cystic fibrosis (PwCF). However, it could have been clearer in presenting a well-defined hypothesis or research question to guide the study, thus strengthening the introduction and indicating specific study objectives to the readers.

Response: The text was revised, as recommended: in Page 2 Line 87-89: “We review the present knowledge of how SARS-CoV-2 influences the release and content of host EVs and how the EVs modulate COVID-19 severity PwCF.”

  1. The text does not include a "conclusions" section. This section could summarize the main findings, discuss their implications, and highlight any recommendations for future research. Adding it would improve the coherence and completeness of the work.

Response:  We have added conclusion section and text was revised, as recommended: Page 6 Line 242-255  “7. Conclusions

In summary, EVs play a significant role in SARS-CoV-2 pathogenesis, with their number increasing during infection and potentially carrying live virus particles [40]. EVs modulate the severity of SARS-CoV-2 infection in PwCF by modifying cytokines expression (Figure 1). EVs can modulate the immune response, acting as decoys to prevent SARS-CoV-2 infection by contesting with cellular ACE2 receptors, and engineered ACE2-loaded EVs are effective against SARS-CoV-2 infection in mice. TGF-β-induced EVs from cancer cells play a critical role in vascular destabilization. These EVs, secreted during EMT, target endothelial cells and contribute to immune suppression and fibrogenesis. Inflammation is a significant component of CF pathogenesis and SARS-CoV-2 infection [112]. The content of EVs in PwCF and SARS-CoV-2 infection is summarized in Figure 1. For instance, EVs derived from the sputum of patients infected with SARS-CoV-2 have been found to express high levels of specific immune-related proteins, which correlate strongly with the expression of the SARS-CoV-2 N protein [113].”

  1. To expedite the reading process of the manuscript, include a graphical abstract.

Response: We provided Figure 1 which act as a graphical abstract. Would you recommend another figure to be included?

  1. Improve the resolution of all figures.

Response: We improved the resolution of Figure 1 from 300 dpi to 600 dpi

  1. The English is understandable and well-structured, and the grammar is generally correct. However, there are some minor corrections that could be made to improve the fluency and clarity of the text. Some sentences could be rephrased to make the meaning more precise or to avoid repetitions, such as lines "22-23" "The COVID-19 pandemic has afflicted more than 773 million people and killed more than 6.9 million individuals between December 2019 and 2023," which could be rephrased: "The COVID-19 pandemic has affected over 773 million individuals and resulted in the deaths of more than 6.9 million people between December 2019 and 2023." These rephrasings could make the text clearer and avoid excessive use of similar words. Additionally, there are some punctuation errors and sentence structure issues that could be corrected to make the text more coherent. Overall, it's quite good, but there's always room for improvement.

Response: The text was revised, as recommended: Page 1 Line 22-24 “The COVID-19 pandemic has affected over 773 million individuals and resulted in the deaths of more than 6.9 million people between December 2019 and 2023.”

Reviewer 2 Report

Comments and Suggestions for Authors

The review by Hejenkowska and colleagues described the role extracellular vesicles in COVID-19 for people with cystic fibrosis. The review is very interesting and well written. The subject matter is quite new and adds something to the current literature.

However, there are areas where the paper could be improved by clarity and dept.

Lines 44-51: I would suggest adding some information about the possible role of the CFTR modulators therapy in SARS-CoV-2 infection in PwCF.

Ref. 27: This is not a peer-reviewed article. Be careful about making statements based on a work as such. Is it reliable?

Line 62: I suggest adding some information on the role of CFTR function/expression in SARS-CoV-2 infection. Are there papers that study SARS-CoV-2 infection by emulating the CF condition by inhibiting CFTR function/expression in vitro (e.g. CFTR KO by CRISPR-Cas or molecular inhibitors)?

Line 65-68: I would suggest deepening the role of extracellular vesicles in host-pathogen interactions with a focus on respiratory viruses.

Line 117-118: I would suggest removing the sentence “Moreover, EVs produced in the airway of PwCF have shown varied protein expression patterns and release more EVs, compared to healthy controls” since you are going to describe this concept in the following paragraph. Instead, I suggest deepening the mechanisms by which EVs modulate host response.

Line 186: correct “SARS-Cov-2” in “SARS-CoV-2”

In paragraph 6 I would suggest contextualizing better the role of EVs in TGF-β signaling in the case of SARS-CoV-2 infection.

I suggest adding a comprehensive table or figure highlighting the differences in EVs between normal conditions and CF conditions.

Author Response

  1. Lines 44-51: I would suggest adding some information about the possible role of the CFTR modulators therapy in SARS-CoV-2 infection in PwCF.

Response: The text was revised, as recommended: Page 2 Line 51-55: “CFTR modulators, a new class of drugs increasing CFTR function in PwCF, decreased SARS-CoV-2 replication in a primary CF airway cell model [25]. CFTR modulators were also associated with a significant decrease in hospitalization of PwCF requiring oxygen and it was concluded that having lower lung function is linked to more severe outcomes in COVID-19 [26].”

  1. 27: This is not a peer-reviewed article. Be careful about making statements based on a work as such. Is it reliable?

Response: The reference in question has been removed and conclusion has been modified based on peer-reviewed data Page 2 Line 68-74 “Some mechanistic cues about the role of CFTR in SARS-COV-2 infection were provided by the following studies. First, CFTR was shown to colocalize with the SARS-CoV-2 receptor, ACE2 in several types of epithelial cells, including those in the respiratory tract [31]. Second, the SARS-CoV-2 nucleocapsid was proposed to downregulate CFTR expression [32]. These findings suggest that the nucleocapsid-dependent loss of CFTR function may result from ACE2 interaction with CFTR, thus initiating a CF-like phenotype and inducing an inflammatory response similar to one experienced by PwCF.”

  1. Line 62: I suggest adding some information on the role of CFTR function/expression in SARS-CoV-2 infection. Are there papers that study SARS-CoV-2 infection by emulating the CF condition by inhibiting CFTR function/expression in vitro (e.g. CFTR KO by CRISPR-Cas or molecular inhibitors)?

Response: The text was revised, as recommended: Page 2 Line 57-67 : “Another study suggested that CFTR may facilitate the virus replication because IOWH-032, a small molecule CFTR inhibitor, suppressed SARS-CoV-2 replication [29]. IOWH-032 is a synthetic extracellular CFTR inhibitor that entered a phase II clinical trial in 2013 to treat diarrhea but has not progressed to clinical development. It was later found to have a potentiating effect on human CFTR [30]. Based on the newer information that IOWH-032 is an ortholog-specific CFTR inhibitor and potentiator, it cannot be concluded that CFTR aids in Sars-CoV-2 replication. Overall, the studies discussed above do not offer a consensus on whether CFTR dysfunction or CF-directed therapies modify the risk of SARS-CoV-2 infection or the severity of COVID-19 in PwCF. It is unknown if CFTR modulators or CFTR expression or function influence SARS-CoV-2 viral load.”

  1. Line 65-68: I would suggest deepening the role of extracellular vesicles in host-pathogen interactions with a focus on respiratory viruses.

Response: The text was revised, as recommended: Page 2 Line 83-87 :  “Respiratory viruses use the host EVs to modulate their transmission. Rhinoviruses induce the release of EVs from airway epithelial cells that stimulate viral receptor expression on monocytes and turn uninfected cells into more permissive [43, 44]. Influenza H5N1 virus induces the release of EVs from infected cells that trigger inflammation [45].”

  1. Line 117-118: I would suggest removing the sentence “Moreover, EVs produced in the airway of PwCF have shown varied protein expression patterns and release more EVs, compared to healthy controls” since you are going to describe this concept in the following paragraph. Instead, I suggest deepening the mechanisms by which EVs modulate host response.

Response: The text was revised, as recommended: Page 3 Line 138-143: “EVs have been shown to modulate the host response to different viruses, including SARS-CoV-2 [74, 75]. For example breast milk derived EVs are able to inhibit HIV-1 infection of monocyte-derived dendritic cells and block viral transfer to CD4+ T cells [76]. EVs have been shown to decrease COVID-19 severity. Blocking proinflammatory cytokine IL-6 and upregulating the anti-inflammatory IL-10  reduces viral replication and decreases systemic inflammation associated with COVID-19 [74, 77, 78].”

  1. Line 186: correct “SARS-Cov-2” in “SARS-CoV-2”

Response: The text was revised, as recommended: Page 4 Line 181 “SARS-CoV-2”

  1. In paragraph 6 I would suggest contextualizing better the role of EVs in TGF-β signaling in the case of SARS-CoV-2 infection.

Response: The text was revised, as recommended: Page 6 Line 224-231 “During SARS-CoV-2 infection, TGF-β signaling plays a crucial role and is particularly prominent in the chronic immune response observed in severe COVID-19 cases [103]. Locally produced TGF-β may contribute to pulmonary fibrosis during the later stages of the immune response [103-106]. Additionally, in PwCF who are homozygous for the most common CFTR gene mutation F508del, the TGF-β gene has been confirmed as a modifier of lung disease severity. Specifically, certain polymorphisms linked to elevated TGF-β1 expression are associated with more severe lung disease and may impact up to 40% of F508del homozygous patients [107-109].”

  1. I suggest adding a comprehensive table or figure highlighting the differences in EVs between normal conditions and CF conditions.

Response: Table highlighting differences in EVs cargo has been included as recommended: Page 4